# Integrated Analysis of mRNA and microRNA Elucidates the Regulation of Glycyrrhizic Acid Biosynthesis in *Glycyrrhiza uralensis* Fisch

**DOI:** 10.3390/ijms21093101

**Published:** 2020-04-28

**Authors:** Yanni Li, Caixia Chen, Zhenzhen Xie, Jing Xu, Bin Wu, Wenquan Wang

**Affiliations:** 1Key Laboratory of Bioactive Substances and Resources Utilization of Chinese Herbal Medicine, Ministry of Education, Institute of Medicinal Plant Development, Chinese Academy of Medical Sciences and Peking Union Medical College, Beijing 100193, China; 2Laboratory of Plant Tissue Culture Technology of Haidian District, Beijing 100091, China

**Keywords:** *Glycyrrhiza uralensis* Fisch, transcriptome, microRNA, glycyrrhizic acid biosynthesis, regulatory proteins, transporter

## Abstract

Licorice (*Glycyrrhiza*) is a staple Chinese herbal medicine in which the primary bioactive compound is glycyrrhizic acid (GA), which has important pharmacological functions. To date, the structural genes involved in GA biosynthesis have been identified. However, the regulation of these genes in *G. uralensis* has not been elucidated. In this study, we performed a comprehensive analysis based on the transcriptome and small RNAome by high-throughput sequencing. In total, we identified 18 structural GA genes and 3924 transporter genes. We identified genes encoding 2374 transporters, 1040 transcription factors (TFs), 262 transcriptional regulators (TRs) and 689 protein kinases (PKs), which were coexpressed with at least one structural gene. We also identified 50,970 alternative splicing (AS) events, in which 17 structural genes exhibited AS. Finally, we also determined that miRNAs potentially targeted 4 structural genes, and 318, 8, and 218 miRNAs potentially regulated 150 TFs, 34 TRs, and 88 PKs, respectively, related to GA. Overall, the results of this study helped to elucidate the gene expression and regulation of GA biosynthesis in *G. uralensis*, provided a theoretical basis for the synthesis of GA via synthetic biology, and laid a foundation for the cultivation of new varieties of licorice with high GA content.

## 1. Introduction

Licorice (*Glycyrrhiza*) plants are the most important dual-use plants and have significant value in ecological protection. The roots and rhizomes of *Glycyrrhiza*, Gancao in Chinese, were first recorded in the *Shengnong Bencao Jing* as a Chinese herbal medicine and natural sweetener. In the Chinese Pharmacopoeia, *Glycyrrhiza uralensis* Fisch, *Glycyrrhiza glabra* L., and *Glycyrrhiza inflata* Bat are the source plants of Gancao [1], in which *G. uralensis* is recognized as the most important species according to the findings of ancient books and modern research [2]. The primary bioactive compound in Gancao is glycyrrhizic acid (GA), which has diverse pharmacological actions, such as anti-inflammatory, anti-cancer, anti-allergic, and anti-viral activities [3]. In addition to medicinal applications, Gancao and its extracts are also widely used in food, cosmetics, and other industries. At present, more than 3000 varieties of medicines and health products were developed using *G. uralensis* and its extracts (http://app1.sfda.gov.cn/datasearchcnda/face3/dir.html). Therefore, the demand for licorice in domestic and foreign markets is considerable. However, the supply of wild *G. uralensis* is limited. The extraction of GA from artificially cultivated *G. uralensis* is time-consuming. Moreover, in recent years, the quality of artificially cultivated *G. uralensis* was not determined to be stable because of the impure germplasm in production, hindering efforts to meet the standards of the Chinese Pharmacopoeia. To solve the problems of sustainable development of *G. uralensis* resources and GA shortage, the GA biosynthesis pathway has been heavily researched in recent years [4]. 

With the release of the *G. uralensis* genome [5], the structural genes involved in GA biosynthesis have gradually been identified. GA belongs to the triterpenoids, and its synthesis is achieved via the mevalonate pathway, which can be classified into two stages: formation of the triterpenoid skeleton and modification of the skeleton [6]. For the first stage, 3-hydroxy-3-methylglutaryl-coenzyme A reductase (HMGR) is the first rate-limiting enzyme that catalyzes the transformation of HMG-CoA to MVA [7]. Squalene synthase (SQS), a key enzyme for the flow of carbon to triterpenes, catalyzes the synthesis of FPP to squalene [8]. Then, β-AS transforms 2,3-oxidosqualene to β-amyrin by two consecutive oxidation reactions [9]. In the second stage, CYP88D6 catalyzes the oxidation of β-amyrin to form 11-oxo-β-amyrin, and CYP72A154 oxidizes the 11-oxo-β-amyrin at the C-30 position to form glycyrrhetinic acid [10]. The cytochrome P450 reductase (GuCPR) is indispensable in the process of C-11 and C-30 oxidation to balance the redox environment in the cell and facilitate the synthesis of glycyrrhetinic acid. Finally, *G. uralensis* Fisch UDP-glycosyltransferase (*Gu*UGT3) catalyzes the glycosylation of glycyrrhetinic acid to form GA [11]. 

However, the regulation of GA biosynthesis in *G. uralensis* has not been elucidated to date. In recent years, regulatory proteins and microRNAs (miRNAs) have received increasing attention. To the best of our knowledge, there are three types of regulatory proteins. Transcription factors (TFs) are regulatory proteins that play important roles in plant growth and development and the biosynthesis of secondary metabolites [12]. TFs could bind to specific motifs in the promoter region of genes such that these genes are upregulated or downregulated [13]. TFs can simultaneously regulate the expression of multiple structural genes in the same biosynthetic pathway. For example, *GmMYB39* inhibited isoflavone biosynthesis by repressing the transcript levels of phenylalanine ammonia-lyase (*PAL*), cinnamate 4–hydroxylase (*C4H*), chalcone synthase (*CHS*), 4-coumarate: coenzyme A ligase (*4CL*) and chalcone reductase (*CHR*) in soybean [14]. Transcriptional regulators (TRs) are also regulatory proteins that can regulate the expression of their target genes via epigenetic modifications, such as H3K14/K9 acetylation [15], trimethylation of H3K4 [16] and H3K27 [17]. TRs were involved in multiple processes, including core trichome initiation [15], auxin response [18], vernalization [17], chlorophyll biosynthesis [16], leaf senescence [19] and plant immunity [20]. Another regulatory protein, protein kinases (PKs), function in signal transduction pathways and alter the activity of target proteins by phosphorylation [21]. PKs played an important role in plant growth [22], abiotic and biotic stress responses, ABA synthesis [23], activation of the transcription factor [24], and secondary metabolism. For example, in *Arabidopsis*, SNF1-related protein kinase 2 (*SnRK2*) and phosphatase 2C (*PP2C*) are up- and down-regulated after UV-B treatment, respectively, which activated the expression of *CHS* and isoflavone synthase (*IFS*), leading to the accumulation of isoflavone [25]. In soybeans, overexpression of the *Arabidopsis thaliana* calcium-dependent protein kinase 1 (*AtCPK1*) gene upregulated the expression of the *4CL*, *IFS*, hydroxyisoflavanone dehydratase (*HID*), dimethylallyltransferase (*IDMAT*), and coumestrol 4-dimethylallyltransferase (*C4-DMAT*) genes, resulting in the increase in isoflavone aglycones [26]. 

Plant miRNAs are nonprotein-coding RNAs measuring approximately 21–24 nt, which are usually derived from a primary transcript with a stem-loop structure. After the action of DICER-LIKE 1 (DCL1), SERRATE (SE), HYPONASTIC LEAVES 1 (HYL1), HUA ENHANCER 1 (HEN1), HASTY (HST), ARGONAUTE 1 (AGO1), STABILIZED 1 (STA1) and RNA binding protein. Mature miRNAs are produced [27,28,29]. These RNAs negatively regulated target genes at the posttranscriptional level by cleavage [30] or translational inhibition [31] or by methylation modification at the transcriptional level [32]. Studies have shown that miRNAs can negatively regulate their target genes to regulate the development of different organs, response to stress [33], and involvement in secondary metabolism [34].

In this study, because roots and leaves in *G. uralensis* have different GA content, we performed transcriptomics and small RNAomics analysis by using these two tissues. We found that TFs, TRs, and PKs genes were coexpressed with multiple structural genes, which were potentially involved in the biosynthesis of GA. We also found that miRNAs potentially regulated the biosynthesis of GA in two aspects. On the one hand, miRNAs potentially targeted structural genes. On the other hand, miRNAs potentially targeted GA-related genes encoding regulatory proteins, including TFs, TRs, and PKs. 

Therefore, a study on miRNAs, TFs, TRs, and PKs would help to reveal the regulatory mechanism of GA synthesis, providing a theoretical basis for the synthesis of GA via synthetic biology and laying a foundation for the cultivation of new licorice varieties with a high GA content. 

## 2. Results

### 2.1. Construction of RNA-Seq Libraries and Sequence Alignment

Our experiment aimed to elucidate the structural genes involved in the GA biosynthetic pathway and their regulation. Because the content of GA is higher in roots than in leaves in *G. uralensis*, we performed transcriptome sequencing on these two tissues and analyzed the differentially expressed genes (DEGs). First, strand-specific cDNA libraries from the poly(A) mRNAs of leaves and roots samples were constructed and subjected to sequence by the Illumina platform. After sequencing and quality control, a total of 45.65 Gb of clean data were obtained, and the percentage of Q30 base in each sample was more than 93.08% (Appendix A). 

Then, we used the draft genome of *G. uralensis* as a reference for sequence alignment and subsequent analysis. After the alignment of the clean reads by HISAT2 [35], we used StringTie [36] to compare and assemble the reads to build the longest transcripts and estimate the expression level. The alignment ratio of reads to reference genomes of each sample ranged from 85.72 to 90.58% (Appendix A).

### 2.2. Gene Function Annotation

To determine gene functions, the longest transcripts were compared with the public databases. As a result, 9827, 22,348, 11,768, 16,758, 22,594, 22,762, 29,118 and 31,910 genes were annotated by Cluster of Orthologous Groups of proteins (COG), Gene Ontology (GO), Kyoto Encyclopedia of Genes and Genomes (KEGG), euKaryotic Ortholog Groups (KOG), Protein family (Pfam), Swiss-Prot protein sequence (Swiss-Prot), Nonsupervised Orthologous Groups (eggNOG), and NCBI nonredundant protein sequences (NR) databases, respectively. We also found a total of 1378 new genes by filtering out sequences encoding peptide chains that were too short (less than 50 amino acid residues) or containing only a single exon (Appendix A).

### 2.3. Structural and Transporter Genes Potentially Involved in GA Biosynthesis

To explore the structural genes involved in the GA pathway, we screened the gene annotations based on the NR and KEGG. The results showed that the 3-hydroxy-3-methylglutaryl-coenzyme A reductase (HMGR) gene contains four transcripts; squalene synthase (SQS), CYP88D6 (C-11 oxidase), and CYP72A154 (C-30 oxidase) genes contain two transcripts; acetyl-CoA C-acetyltransferase (AACT), 3-hydroxy-3-methylglutaryl coenzyme A synthase (HMGS), mevalonate kinase (MK), phosphomevalonate kinase (PMK), mevalonate diphosphate decarboxylase (MPD), farnesyl diphosphate synthase (FPPS), beta-amyrin synthase (β-AS), and CPR reductase (GUCPR) genes contain one transcript. The transcript information of these structural genes is shown (Figure 1, Table 1). Unfortunately, the structural genes encoding isopentenyl diphosphate isomerase (IPI), squalene epoxidase (SE), and *Glycyrrhiza uralensis* Fisch UDP-glycosyltransferase (*Gu*UGT 3) were not found in our RNA-Seq data, indicating the low expression level of these genes in our samples. In addition to structural genes, we also focused on the transporter genes because the enzymes, encoded by structural genes, that synthesize the intermediate products of GA are localized in different organelles (Table 1). Through a comparison with the TCDB transporter database, a total of 3924 transporter candidate genes were identified in *G. uralensis* (Appendix A).

### 2.4. Identification of Regulatory Proteins in G. uralensis

It was reported that regulatory proteins, including TFs, TRs, and PKs, play important roles in plant growth and development and the synthesis of secondary metabolites. To explore the roles of these proteins in *G. uralensis*, we first predicted these regulatory proteins by comparing the nucleotide or coding-protein sequences of the longest transcripts with the iTAK database. In total, we found 1841 TFs belonging to 69 families, in which the number of MYB family TFs was highest (Appendix A). We identified a total of 401 TRs belonging to 23 families (Appendix A). We also identified 1168 PKs, which belonged to 119 families. Among these PKs, the RLK-Pelle_DLSV group was the largest family (Appendix A). 

### 2.5. Identification of DEGs and qRT-PCR Validation

Compared with roots, those genes that had an expression level of |fold change| > 2 and FDR < 0.01 in leaves found by StringTie were assigned as DGEs. In total, we obtained 4284 DEGs, including the GA pathway structural genes *HMGS*(Glyur000195s00012841), *HMGR*(Glyur000037s00002618)/ (Glyur000203s00012900)/ (Glyur000682s00024324) and *SQS* (Glyur000089s00008825), 680 transporters, 321 TFs, 43 TRs, and 213 PKs (Appendix A).

To verify the reliability of RNA-Seq results, 12 structural genes, including *AACT* (Glyur000218s00011642.1), *HMGS*(Glyur000195s00012841.1), *HMGR*(Glyur000682s00024324.1), MK(Glyur000069s00004081.1), *PMK*(Glyur000343s00025703.1), *MPD*(Glyur000002s00000233.1), *FPPS*, (Glyur000088s00007722.1), *SQS1*(Glyur000089s00008825.1), *β-AS* (Glyur001733s00027628.1), *CYP88D6*(Glyur000561s00023451.1), *CYP72A154*(Glyur000890s00019071.1), and *GUCPR1* (Glyur000294s00011848.1) were selected for qRT-PCR validation. The results of qRT-PCR were highly consistent with the RNA-Seq data, and the correlation coefficient was 0.9525 (Figure 2).

### 2.6. GO Enrichment Analysis of DEGs

The GO database describes gene function across species. The results showed that a total of 3219 DEGs were annotated by GO (Appendix A). The top 50 secondary subclassifications are listed in Figure 3, including biological process (20), cellular component (16) and molecular function (14). The top five of the DEGs are involved in catalytic activity (1852), metabolic process (1752), cellular process (1436), binding (1411) and cell (1360), which showed the same tendency with all the GO annotated genes (Figure 3).

### 2.7. KEGG Enrichment Analysis of DEGs

KEGG allows for functional annotations of gene products according to various metabolic pathways. Figure 4A shows the top 50 subcategories, including cellular processes (38), environmental information processing (46), and genetic information processing (59). Interestingly, many DEGs are involved in metabolism (1047) and organismal systems (42) (Figure 4A, Appendix A).

The results of the KEGG pathway enrichment analysis of DEGs are shown in Figure 4B, which shows the top 20 pathways with the Q value. We found that some genes are enriched in terpenoid synthesis pathways and metabolism processes, in which 32, 23, and 7 genes are enriched in ubiquinone and other terpenoid-quinone biosynthesis (7 transporters), terpenoid backbone biosynthesis (4 transporters), and sesquiterpenoid and triterpenoid biosynthesis, respectively (Figure 4B).

### 2.8. Correlation Analysis between Transporter and Structural Genes

Of the 3924 transporter genes, 3556 have expression levels (Appendix A). Then, we calculated the Pearson correlation coefficients (r) of the expression profiles between transporter genes and the GA structural genes. With the threshold of | r | > 0.8, a total of 2374 transporter genes were identified. Among these genes, 713, 435, 161, 66, 41, 41, 136, 311, and 1077 transporter genes were found to be highly correlated with 10, 9, 8, 7, 6, 5, 4, 3, 2, and 1 structural gene(s), respectively (Appendix A). Subsequently, subcellular localization prediction of the 2374 transporters was carried out by ngLOC (http://genome.unmc.edu/ngLOC/index.html) (Appendix A), in which 53 transporters predicted to be localized in the vacuole and 23 transporters to be localized in the endoplasmic reticulum (ER) (Appendix A), consistent with the location of most of the GA metabolic enzymes. This information strongly enhances our knowledge of transporters involved in GA metabolism.

### 2.9. Correlation Analysis between Regulatory Protein and the Structural Genes

Of the 1841 TFs, 1552 had expression levels (Appendix A). Of the 401 TRs, 368 had expression levels (Appendix A). Of the 1168 PKs, 1034 had expression levels (Appendix A).

To decipher the role of regulatory protein genes in GA metabolism, coexpression analysis was performed to characterize the expression patterns of TF, TR and PK genes and the structural genes. Using a threshold of | r | > 0.8, a total of 1040 TFs were associated with at least one structural gene in the GA metabolism pathway, in which 289, 204, 80, 43, 27, 13, 18, 62, 148, and 445 TFs were highly correlated with 10, 9, 8, 7, 6, 5, 4, 3, 2, and 1 enzyme gene(s), respectively. Among the 1040 TFs, BHLH (82), MYB (74), C2H2 (64), AP2/ERF (60), WRKY (59), bZIP (47), MYB-related (45), NAC (44), C3H (41), HB-HD-ZIP (35) were the top 10 families. To date, six TF families in plants have been reported to be associated with terpenoid metabolism, including AP2/ERF, bHLH, MYB [37], NAC [38], WRKY [39], and bZIP [40]. 

In total, 262 TRs were associated with at least one structural gene in the GA metabolism pathway, in which 52, 29, 15, 5, 6, 6, 5, 35, 68 and 93 TRs were related to 10, 9, 8, 7, 6, 5, 4, 3, 2, and 1 structural gene(s), respectively. Among the 262 TRs, others (43), GNAT (27), SET (26), SNF2 (26), AUX/IAA (25), mTERF (24), PHD (19), TRAF (11), ARID (11), and SWI/SNF-BAF60b (11) were the top 10 family members.

A total of 689 PKs were associated with at least one structural gene in the GA metabolism pathway, in which 188, 132, 59, 26, 15, 8, 14, 38, 98, and 299 PKs were related to 10, 9, 8, 7, 6, 5, 4, 3, 2, and 1 structural gene(s), respectively. Among the 689 PKs, RLK-Pelle_DLSV (50), RLK-Pelle_LRR-XI-1 (45), RLK-Pelle_RLCK-VIIa-2 (29), RLK-Pelle_LRR-III (25), CAMK_CDPK (23), RLK-Pelle_LRK10L-2 (21), CAMKL-CHK1(20), RLK-Pelle_LysM (20), TKL-Pl-4(19), RLK-Pelle_L-LEC (19) were the top 10 family members in PKs (Appendix A). 

All 1040 TFs, 262 TRs, and 689 PKs supplied useful candidate components to enhance GA biosynthesis in *G. uralensis*.

### 2.10. Binding Site Analysis of R2R3-MYB in the Promoter Region of the Structural Gene

The recognition and binding of TF to the target gene promoter is the most critical link in gene expression regulation. In plants, the MYB factor is one of the largest TF families [41]. MYB could be divided into three subfamilies according to the number of adjacent repeats in the MYB domain, in which the R2R3-MYB subfamily, which mainly regulates plant-specific processes [12], was further studied. The current research has verified that the core sequence of the R2R3-MYB binding site is TAACTG [42]. Therefore, we searched it at the promoter regions (2000 bp upstream of the initiation codon) of the structural genes (Table 2). As shown in Appendix A, the promoter regions of *AACT* (Glyur000218s00011642.1), *HMGR* (Glyur000682s00024324.1), *MK* (Glyur000069s00004081.1), *MPD* (Glyur000002s00000233.1), *SQS1* (Glyur000089s00008825.1), and *CYP88D6* (Glyur000561s00023451.1) had binding sites. There are 119 TFs of the MYB family found in *G. uralensis*, of which three R2R3-MYBs (Glyur000618s00030214, Glyur000049s00003134 and Glyur000266s00012631) were related to 10, 10 and 9 structural genes, respectively [37].

### 2.11. AS Analysis

In total, 50,970 AS events were identified by ASprofile, including 2168 exon skipping (ES), 20,126 Alt 3′ (alternative 3′ last exon, TTS), 20,700 Alt 5′ (alternative 5′ first exon, TSS) and 3671 intron retention (IR), respectively [43]. ES could be classified into SKIP (skipped exon), XSKIP (approximate SKIP), MSKIP (multiexon SKIP) and XMSKIP (approximate MSKIP). IR included IR (intron retention), XIR (approximate IR), MIR (multi-IR) and XMIR (approximate MIR). Then, we focused on the AS in the GA biosynthetic structural genes. The results are shown in Table 3. More than one AS event occurred in the 17 structural genes of GA. All the structural genes were AS of the TSS and TTS types. We also found a total of 2694 transporters with AS, including 89 ES, 2655 Alt 3′, 2616 Alt 5′ and 171 IR. A total of 1073 TFs occurred as AS events, including 41 ES, 1065 Alt 3′, 1047 Alt 5′ and 50 IR. A total of 297 TRs occurred with AS, including 12 ES, 294 Alt 3′, 293 Alt 5′, and 35 IR. A total of 775 PK occurred AS, including 42 ES, 763 Alt 3′, 752 Alt 5′, and 74 IR. Most of the transporter, TF, TR and PK genes occurred in the TSS and TTS types of AS (Appendix A).

### 2.12. Features of G. uralensis Small RNA Population

To identify miRNAs in *G. uralensis*, we first constructed small RNA libraries of roots and leaves (each tissue with three replicates). Then, these libraries were sequenced by the HiSeq NovaSeq platform. After sequencing, to collect the clean data, the raw reads were filtered by removing adapters, low-quality sequences, sequences less than 18 nt and sequences larger than 30 nt. Then, the clean data of the six libraries were pooled together to analyze the overall features of the *G. uralensis* sRNA population. In total, 116,807,113 clean reads represented by 20,048,413 unique sRNA sequences were obtained. Among these reads, the group of 24 nt clean reads was the highest, accounting for 19.24% of the total clean reads (Appendix A), and the group of 24 nt unique sRNA sequences was the largest, accounting for 40.43% of the total unique sequences (Appendix A), which was consistent with the findings of previous reports in some plants, such as *Arabidopsis* [44], *Oryza sativa* [45], and *Zea may* [46].

As shown in Appendix A, the groups of 21 nt and 24 nt sRNAs had the highest proportions of U and A nucleotides at the first base, respectively, indicating the different proportions of sRNAs. It was reported that miRNAs and trans-acting small interfering RNAs prefer beginning with a U base, and antisense transcript-derived siRNAs and repeat-associated siRNAs tended to begin with an A base (Appendix A).

### 2.13. Identification and Validation of MiRNAs in G. uralensis

The psRobot software was used to predict the miRNAs in *G. uralensis*. Then, these predicted miRNAs were checked manually according to the criteria of miRNA, as described by Meyer et al. In total, 1894 miRNAs were identified in *G. uralensis*. After comparison with miRbase, it was found that 198 miRNAs were conserved and could be divided into 29 families, of which miR4389 was the most heavily represented family, with 73 members. The remaining 1696 miRNAs were identified as *G. uralensis*-specific miRNAs, which could be divided into 518 families according to the different precursors. Among these miRNAs, the largest family (miR-n4) had 294 members (Appendix A). 

To verify the authenticity of miRNA, 6 miRNAs, including miR172c-3P, miR156a, miR164a-5P, miR156c-3P, miR166-3P and miR2119, which targeted TFs or the *GUCPR1* genes, were verified by stem-loop PCR. The results showed that the melting curves of the six miRNA amplification products were all single peaks, which validated the authenticity of these miRNAs (Appendix A).

### 2.14. Target Prediction of MiRNAs in G. uralensis

MiRNAs could bind and negatively regulate their target genes in a not completely complementary pairing manner [27]. To decipher the function of miRNAs in *G. uralensis*, targets of the miRNAs were predicted by psRNATarget software using an expectation penalty score of ≤3. As a result, we predicted 553 and 2474 targets for the conserved and species-specific miRNAs in *G. uralensis*, respectively. In total, 2866 target genes were obtained, of which 2171 had GO annotations and 1081 had KEGG annotations. A total of 448 target genes were DEGs, of which 374 had GO annotations and 166 genes had KEGG annotations (Appendix A). GO enrichment of target genes showed that the top five classifications were catalytic activity (1137), binding (1089), metabolic process (1050), cellular process (1045), and cell (991). GO terms of the top five classifications of the 374 DEGs were catalytic activity (221), metabolic process (200), membrane (187), binding (168), and cellular process (164) (Figure 5A, Appendix A). 

KEGG enrichment analysis of target genes is shown in Figure 5B. We found that some genes were enriched in terpenoid synthesis pathways and metabolism processes, in which 1, 1, 1, and 3 genes were enriched in diterpenoid biosynthesis, monoterpenoid biosynthesis, sesquiterpenoid and triterpenoid biosynthesis, and terpenoid backbone biosynthesis, respectively (Figure 5B, Appendix A).

One of our concerns was to determine if the structural genes of GA were targeted by miRNAs. As a result, four structural genes were found to be potentially targeted by 12 miRNAs belonging to four families (Table 4). Among them, *MK*(Glyur000069s00004081.1) was potentially targeted by the nine miRNAs of the miRNA-n217 family. miRNA-n244, miRNA-n69 and miR2119 potentially targeted *β-AS*(Glyur001733s00027628.1), *HMGR*(Glyur000682s00024324.1), and *CPR1*(Glyur000294s00011848.1) genes, respectively. The structures of miRNA-n217g, miRNA-n69, miRNA-n244, and miR2119 precursors are shown in Figure 6.

We also investigated GA-related transporter genes which are regulated by miRNAs. The results are presented in Appendix A. We found that 83 conserved and 884 *G. uralensis*-specific miRNAs targeted 53 and 269 transporter genes, respectively, in which 21 transporters were potentially regulated by conserved and *G. uralensis*-specific miRNAs simultaneously. Of these 322 transporters, 9 transporters were localized in the vacuole, and 1 transporter was localized in the ER.

miRNAs might regulate GA synthesis by acting with GA-related regulatory protein genes. The results showed that 223 conserved and 195 *G. uralensis*-specific miRNAs targeted 60 and 90 TF genes, respectively, in which 22 TFs were potentially regulated by conserved and *G. uralensis*-specific miRNAs simultaneously (Appendix A).

It was reported that overexpression of MicroRNA156 could improve drought stress tolerance and biomass yield in Medicago sativa L [47,48]. However, under drought conditions, the synthesis of GA was enhanced. Therefore, we further analyzed the target genes of miRNA156 in *G. uralensis*. In total, 10 members of miRNA156 were identified, potentially targeting 15 potential GA-related TFs, including 12 *SPL*s (Appendix A), in which *SPL3, SPL 6, SPL12* and *SPL13* cleavage by MicroRNA156 was reported in Salvia miltiorrhiza and Medicago sativa L [48,49].

For TRs, we found 6 conserved and 74 *G. uralensis*-specific miRNAs acted on 60 and 90 TR genes, respectively, in which 2 TRs were potentially regulated by conserved and *G. uralensis*-specific miRNAs simultaneously (Appendix A). We also found 31 conserved and 187 *G. uralensis*-specific miRNAs potentially acted on 16 and 83 PKs genes, respectively, in which 5 PKs were potentially regulated by conserved and *G. uralensis*-specific miRNAs at the same time (Appendix A).

### 2.15. Negative Correlation Expression between MiRNA and Target Transcripts

We identified a total of 1894 miRNAs in *G. uralensis*, of which 1726 had expression levels (Appendix A). Using the threshold of |log_2_ fold change| ≥ 2 and FDR < 0.05 found by DESeq 2 package, we identified a total of 35 DE miRNAs. Among these miRNAs, 29 were downregulated, and 6 were upregulated (Appendix A). Then, we focused on the expression patterns of negative correlations between miRNAs and their target genes. As a result, six miRNA/protein-coding gene pairs were identified (Figure 7). This group included miR156f/*SPL6* (Glyur000050s00005683), GumiRNA-n298/Putative transporter arsB (Glyur000323s00028436), GumiRNA-n315/phylloquinone biosynthesis protein (Glyur000239s00017890), miR390a/b-5P/hypothetical protein LR48_Vigan09g172300 (Glyur000005s00001047), miR5037a-5P/pentatricopeptide repeat-containing protein and chloroplastic (Glyur000247s00015505), and GumiRNA-n194-3p/hypothetical protein KK1_020744 (Glyur000039s00004183)/chalcone reductase genes (Glyur000746s00026700) (Appendix A, Figure 7).

Then, we focused on the miRNAs/GA-related TFs. For miR156f, one of its target genes (Glyur000050s00005683) was determined to encode *SPL6*. The expression profiles of Glyur000050s00005683 were negatively correlated with nine structural genes (Appendix A).

## 3. Discussion

In this study, we performed transcriptional analysis to reveal the expression and regulation of GA biosynthesis in *G. uralensis*. For the structural genes of GA, three aspects should be considered. First, some family genes contain more than one locus in the genome. To understand the gene function comprehensively, it is necessary to study all members because of the possibility of functional diversity for different members. For example, diterpene syntheses genes have different functions in *Salvia miltiorrhiza* [50]. Taking the cellulose synthase (*CESA*) genes as another example, there are 10 *CESA* genes in *Arabidopsis* [51]. The *irx1*, *irx3*, and *irx5* mutants of *Arabidopsis* are caused by mutations in the *AtCESA7*, *AtCESA8*, and *AtCESA4* genes, respectively; the encoding proteins of these three genes also interact with one another and participate in the cellulose synthesis of the secondary cell wall [52]. Second, AS could enhance the protein diversity of genes [53]. In medicinal plants, such as *Digitalis purpurea* and *Salvia miltiorrhiza*, AS occurred in genes involved in the biosynthesis of cardiac glycosides, and tanshinone was deciphered [53,54]. Moreover, it is of significance to study the polymorphisms of structural genes, such as SNPs or inserts/indels, in different cultivars, subspecies or mutants, which could lead to the change of enzyme activity. For example, in *Oryza Sativa*, the *gh2* mutant is a lignin-deficient mutant, and gold hull and internode2 (*GH2*) encodes a cinnamyl-alcohol dehydrogenase [55].

Transporters function in the transportation of various substrates, such as anions, sugars/maltose, amino acids, proteins, mRNAs, electrons, and hormones, which play an important role in biological processes [56]. In this study, we found that 2374 transporter genes were coexpressed with at least structural genes, of which 53 transporters were predicted to be in the vacuole and 23 transporters to be localized in the ER. Because most of the enzyme of GA is localized in the ER, our results indicate that the 23 transporters are potentially involved in the transportation metabolic intermediate of GA. Further experiments may confirm this possibility.

To date, regulatory protein genes involved in the biosynthesis of GA have not been reported. In this study, we found that 1040 TFs, 262 TRs, and 689 PKs genes were correlated with the structural genes of GA for the first time. Previously, TFs have been shown to be involved in terpenoid metabolism. For example, MsMYB, a R2R3-MYB TF, repressed monoterpene production [37]. We also found that TFs of AP2/ERF, bHLH, MYB, NAC, WRKY, and bZIP were potentially associated with terpenoid metabolism, consistent with the findings of a previous report. Among these TFs, R2R3-MYBs, including Glyur000618s00030214, Glyur000049s00003134, and Glyur000266s00 012631, were coexpressed with 10, 10, and 9 structural genes, respectively. Then, we further analyzed the binding site (TAACTG) of R2R3-MYB in the promoter region of the structural genes of GA. Fortunately, this site was found in the promoter regions of six structural genes (Table 2). Therefore, the three R2R3-MYBs were highly likely to interact with these structural genes. To date, no TRs have been reported to be involved in secondary metabolism. For PKs, it is reported that PKs are involved in the biosynthesis of isoflavone [25,26]. The discovery of GA-related TFs, TRs, and PKs supply candidate components for synthetic biology. 

miRNAs may regulate the biosynthesis of GA in two aspects. On the one hand, some miRNAs potentially target structural genes, that is, genes encoding regulatory proteins positively related to GA synthesis (Figure 1). To cultivate a new variety of *G. uralensis* with higher GA content, it is necessary to knock out these miRNAs. On the other hand, other miRNAs may target genes encoding regulatory proteins negatively related to GA synthesis. Overexpression of these miRNAs might enhance GA biosynthesis in *G. uralensis*. For the complex regulation of miR156, this miRNA potentially targets 15 TFs, 7 and 5 of which were negatively and positively related to the structural genes of GA, respectively, indicating that GA synthesis may be enhanced by the overexpression of miR156. Further experiments may validate this possibility.

Taken together, these results helped to characterize gene expression and regulation in *G. uralensis*, provided a theoretical basis for the synthesis of GA via synthetic biology, and laid a foundation for cultivating new licorice varieties with high GA content.

## 4. Materials and Methods 

### 4.1. Plant Materials and RNA Extraction

Two-year-old *G. uralensis* “B11”, wild accession (unpublished) plants were grown in the experimental field of Beijing Medicinal Plant Garden of the Institute of Medicinal Plant Development, Chinese Academy of Medical Sciences and Peking Union Medical College (Beijing, China) in natural growing conditions. Leaves and roots were collected from three plants (one plant for a biological repeat) on September 3, 2018, and frozen in liquid nitrogen. At that time, the temperature ranged from 20 to 32 °C. Total RNA from leaves and roots was extracted using the RNAprep Pure Plant Kit (DP171221) (TIANGEN Biotech (Beijing, China) Co., Ltd.) and TRIzol reagent (Invitrogen, Carlsbad, CA, USA), respectively, according to the manufacturer′s instructions. Then, RNAs were assessed and quantified as described previously [57]. 

### 4.2. Illumina Transcriptome Library Construction and Sequencing

The poly(A) mRNAs of leaf and root samples (every organ with three replicates) were separated from the total RNA using oligo(dT) magnetic beads (Dynal, Oslo). Then, the mRNAs were used to construct strand-specific transcriptome libraries, as in our previous work [57]. Finally, these libraries were sequenced using the Illumina platform.

### 4.3. Illumina Data Mapping and Quantification of Gene Expression Levels

Raw reads in FASTQ format were first processed using in-house Perl scripts. Clean reads were obtained by removing reads containing adapters, reads containing poly(N) and low-quality reads by Trimmomatic software (v0.30, Aachen, Germany) [58]. Then, these clean reads were mapped to the reference genome of *G. uralensis* using HISAT (v2, Baltimore, MD, USA) [35] with the default parameter. Subsequently, the mapping reads were assembled, and the expression levels were calculated using StringTie (v1, Baltimore, MD, USA) [36]. Compared with roots, those genes that had a |fold change| ≥ 2 and FDR < 0.01 found by StringTie were assigned as differentially expressed genes (DEGs).

### 4.4. Gene Function Annotation

To obtain annotation information, the longest transcripts were annotated by comparison with the data in the Nr [59], Swiss-Prot [60], GO [61], COG [62], KOG [63], Pfam [64], and KEGG [65] databases, respectively, by BLAST (v2.2.17) [66]. Compared with previous annotations in the draft genome of *G. uralensis*, new genes were acquired. Then, these new genes were compared with the Pfam database (http://pfam.sanger.ac.uk/) using the HMMER software (v3.1b2) [67].

### 4.5. Identification of TF, TR, PK, and Transporter

To predict candidate TFs, TRs, and PKs in *G. uralensis*, nucleotide sequences for TFs and TRs and translated protein sequences for PKs of the longest transcripts were compared to the sequences in the iTAK database (http://bioinfo.bti.cornell.edu/cgi-bin/itak/index.cgi) using the default parameters [21]. To analyze transporters, the protein sequence of the longest transcripts was compared with the sequence in the Transporter Classification Database (TCDB) [68] by BLAST using the default parameter. 

### 4.6. Detection of AS Events

AS events were identified as follows. First, the alignment results of HISAT (v2) [35] were spliced using StringTie software [36], and the alternative splicing type and corresponding expression amount of each sample were obtained using ASprofile software (v2, Baltimore, MD, USA) [43]. 

### 4.7. Validation of the Expression Profile of Transcripts by qRT-PCR

The RNA samples, as described for RNA-Seq library construction, were used for qRT-PCR analyses. First, the RNAs were digested with RNase-free DNase I to remove residual genomic DNA during the RNA extraction process. Then, reverse transcription, qRT-PCR and calculation of gene relative expression levels were performed as previously described in He [57]. The primers are listed in Appendix A. The Pearson correlation coefficient of the expression profiles was also calculated.

### 4.8. Small RNA Library Construction and High-Throughput Sequencing

A small RNA library was prepared as described [69]. Briefly, 10–30 nt small RNAs were purified by a 15% denaturing polyacrylamide gel and then ligated to the adapters. After reverse transcription by M-MLV (TaKaRa, Dalian, China), these small RNAs were amplified by PCR. Finally, high-throughput sequencing was performed using the HiSeq NovaSeq platform.

### 4.9. MiRNA Identification and Target Prediction

After removing adapters and filtering low-quantity sequences by Trimmomatic software [58], the 18–30 nt clean reads of small RNAs in leaves and roots were pooled together, and small RNAs with counts ≥5 were further compared with the draft genome sequences of *G. uralensi* using the precursor predicted module “psRobot-mir” of the psRobot software (v1.2) [70]. The criteria described by Meyers and Zhang were applied to check *G. uralensis* microRNAs manually [71,72]. The minimal folding free energy index (MFEI) was calculated as described previously [73]. To categorize these miRNAs, we first distinguished and named them according to the precursor. Then, for the analysis of conserved miRNAs, the sequences were compared with the sequences in miRbase, allowing three bases of mismatch. To compare the expression level of miRNAs, the count of each miRNA was normalized to transcripts per million (TPM) following the statistical method as described previously. Compared with the expression level in roots, those miRNAs in leaves that had a |log_2_ fold change| ≥ 2 and FDR < 0.05 found by DESeq (v2) [74] were assigned as differentially expressed miRNAs (DEmiRNAs). Targets of the miRNAs were predicted by the psRNATarget Server using a penalty score ≤ 3 [75]. 

### 4.10. Validation of the Authenticity of MiRNAs by Stem-Loop PCR

Approximately 200 ng of DNA-free total RNA was hybridized with a miRNA-specific stem-loop RT primer. Then, the hybridized miRNA molecules were reverse transcribed into cDNA as described [76]. The resulting cDNA was diluted and then PCR-amplified in triplicate using the Bio-Rad CFX96 Real-Time PCR System (Bio-Rad, USA). Each PCR analysis was performed in a final volume of 20 μL containing 10 μL 2x SYBR Premix Ex Taq II (TaKaRa, Dalian, China), 0.25 μM each of the miRNA-specific forward primer and the universal reverse primer (Appendix A), and cDNA reverse-transcribed from 10 ng total RNA was obtained with the following conditions: 95 °C for 20 s, 40 cycles of 95 °C for 10 s, 60 °C for 10 s and 72 °C for 6 s. To verify the specificity of PCR amplification, melting curves were analyzed.

## Figures and Tables

**Figure 1 ijms-21-03101-f001:**
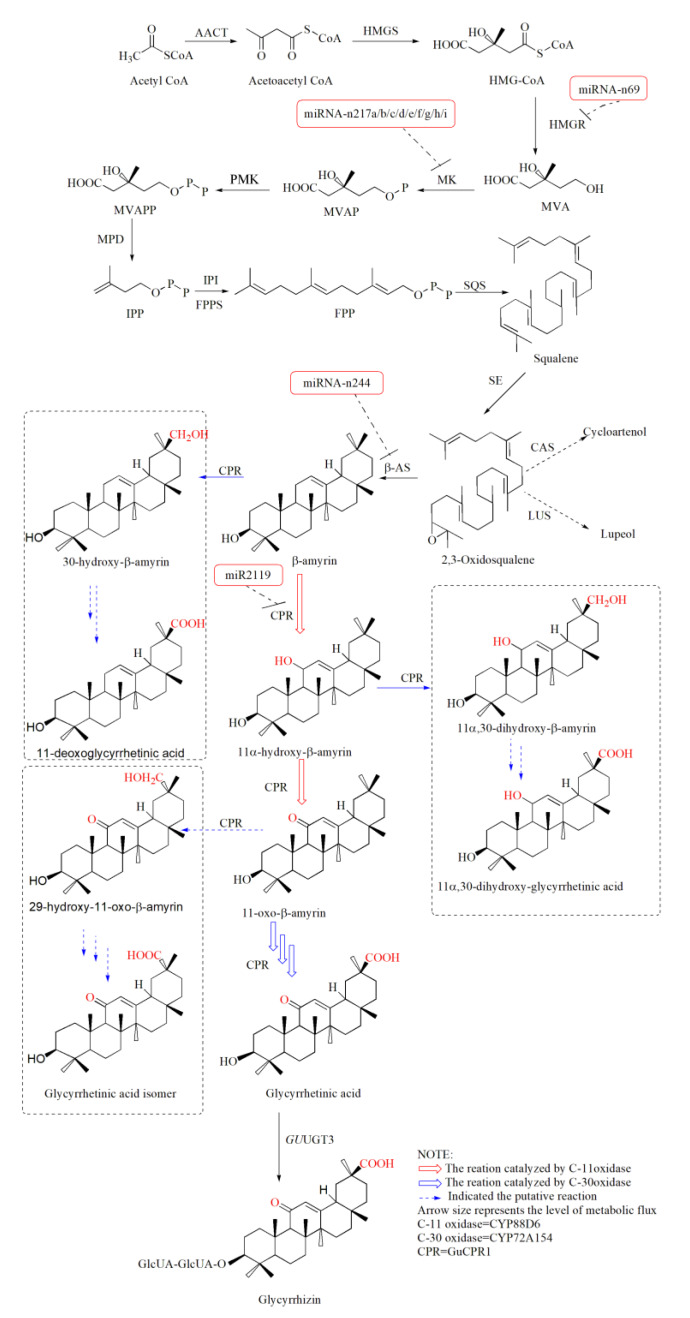
Proposed GA biosynthetic pathway in *G. uralensis*. Red, blue arrows indicate reactions catalyzed by C-11, C-30 oxidase, respectively. Dotted arrows indicate the proposed reactions. Structural genes potentially targeted by the identified *G. uralensis* miRNAs are represented by red rectangular boxes. The dotted line represents uncertain action in the pathway. AACT: acetyl-CoA C-acetyltransferase; HMGS: 3-hydroxy-3-methylglutaryl coenzyme A synthase; HMGR: 3-hydroxy-3-methylglutaryl-coenzyme A reductase; MK: mevalonate kinase; PMK: phosphomevalonate kinase; MPD: mevalonate diphosphate decanmrboxylase; IPI: isopentenyl diphosphate isomerase; FPPS: farnesyl diphosphate synthase; SQS: squalene synthase; SE: squalene epoxidase; β-AS: beta-amyrin synthase; C-11 Oxidase:CYP88D6, beta-amyrin 11-oxidase; C-30 Oxidase: CYP72A154, 11-oxo-beta-amyrin 30-oxidase; CPR: cytochrome P450 reductase; *Gu*UGT3: *G. uralensis* Fisch UDP-glycosyltransferase 3.

**Figure 2 ijms-21-03101-f002:**
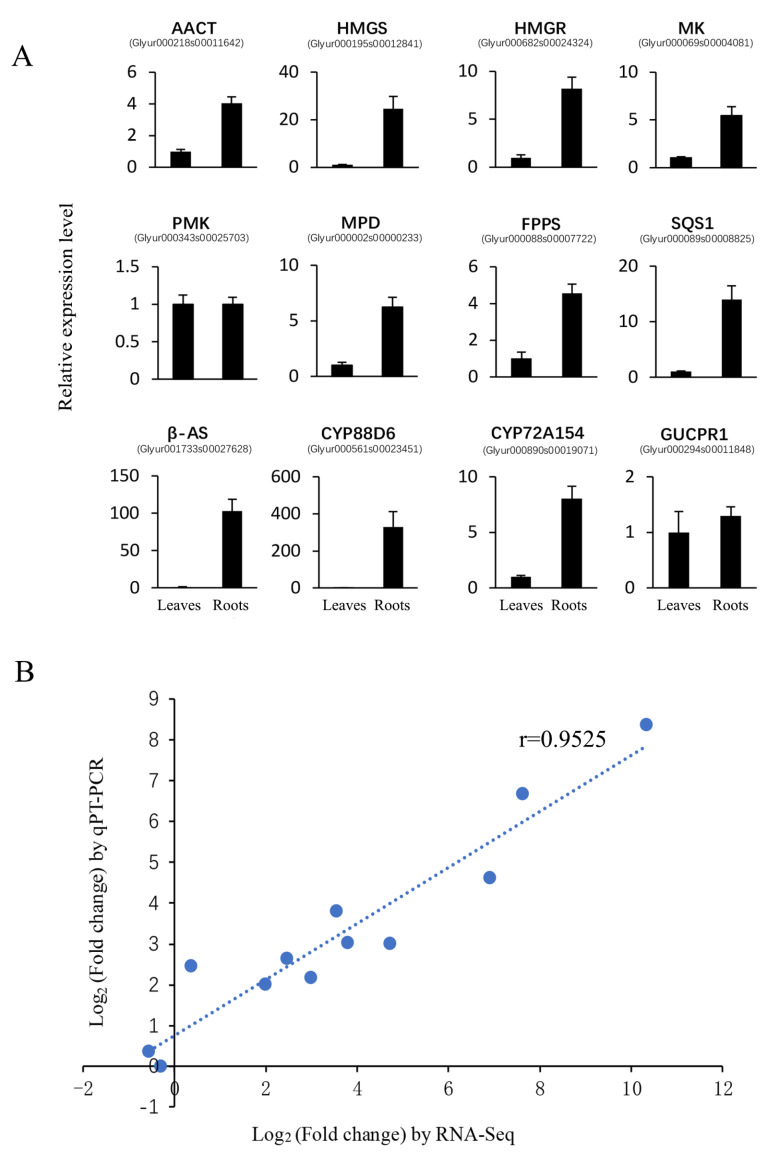
qRT-PCR validation of structural genes (**A**) and the correlation between qPT-PCR and RNA-Seq results (**B**). In Figure 2 (A), the expression level of the structural gene in the leaves was set to 1, the relative expression of genes in roots was calculated, and the error bar is shown. In Figure 2 (B), the *X*-axis represents the log2 (fold change) of the RNA-Seq results, and the *Y*-axis is the log2 (fold change) of the qRT-PCR results.

**Figure 3 ijms-21-03101-f003:**
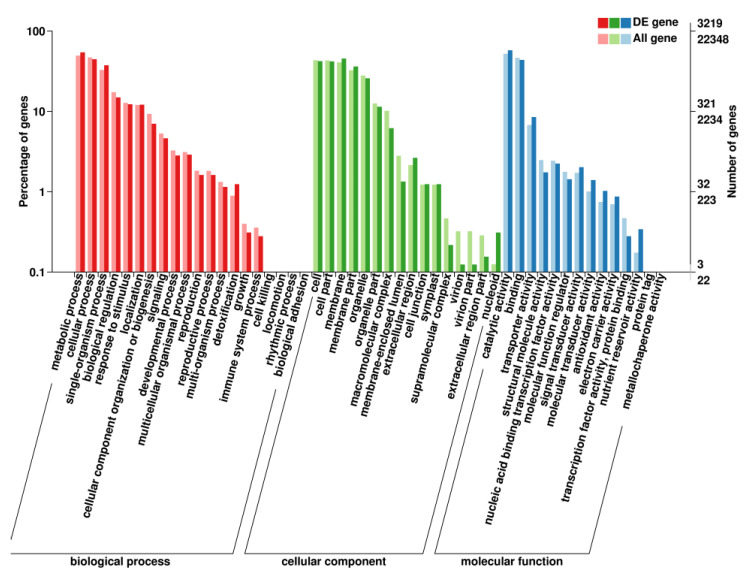
Results of GO enrichment. The percentage of genes assigned to partial subclassifications is shown. Red, green and blue columns represent the biological, cellular, and molecular functions, respectively.

**Figure 4 ijms-21-03101-f004:**
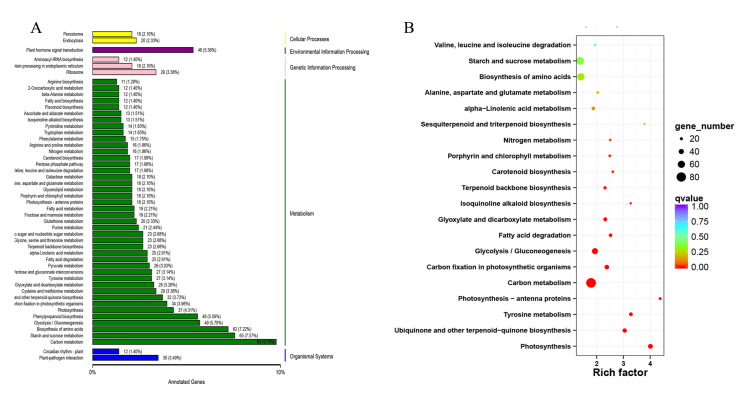
(**A**) KEGG classification of DEGs. The *X*-axis indicates the percentage of the annotation genes, and the *Y*-axis represents the KEGG pathway. (**B**) KEGG pathway enrichment of DEGs for the comparison of roots vs. leaves. The *X*-axis indicates the enrichment factor and log10 of the Q-value. The *Y*-axis represents the different KEGG pathways.

**Figure 5 ijms-21-03101-f005:**
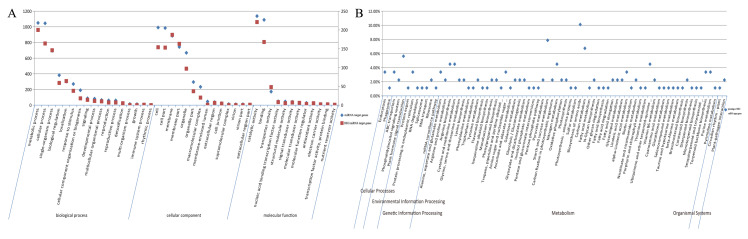
(**A**) GO subclassification of miRNA target genes. (**B**) KEGG subclassification of miRNA target genes.

**Figure 6 ijms-21-03101-f006:**
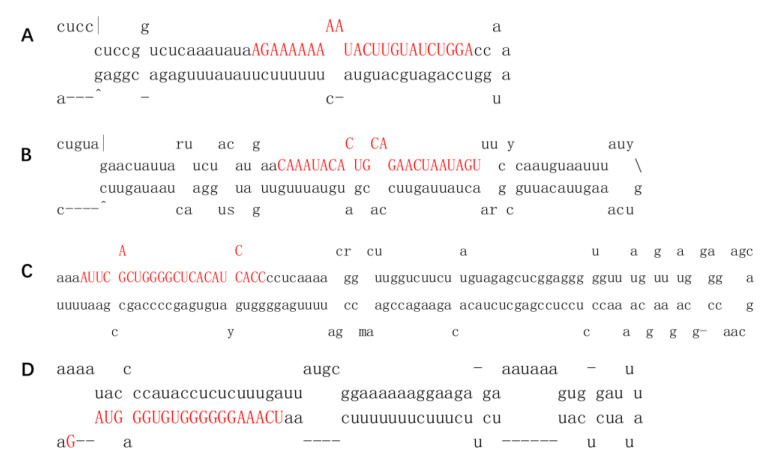
Precursor structure of miRNA-n217g (**A**), miRNA-n244 (**B**), miRNA-n69 (**C**), and miR2119 (**D**). The red capitalized bases represent the sequences of mature miRNA.

**Figure 7 ijms-21-03101-f007:**
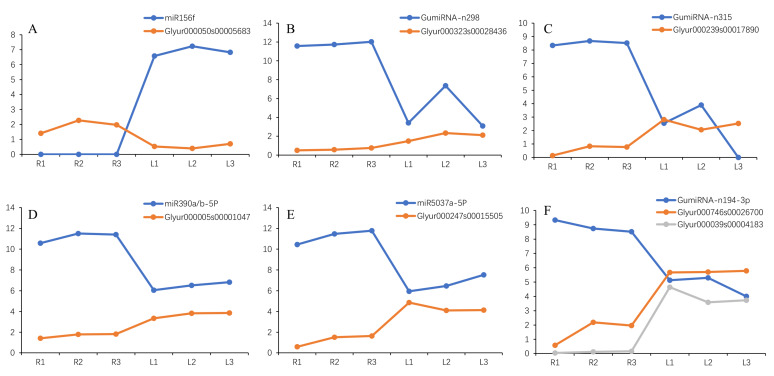
Expression patterns of six miRNAs and target genes. L, R stand for leaves and roots, respectively. Every tissue has three biological replicates.

**Table 1 ijms-21-03101-t001:** Genes potentially involved in the GA biosynthetic pathway.

Annotation	Transcript Number	Transcript ID	Predicted Subcellular Localization	Expression Profiles (FPKM)						Up-Regulated in Roots (Y/N)
				L1	L2	L3	R1	R2	R3	
*AACT*	1	Glyur000218s00011642.1	Peroxisome	38.32	42.15	47.59	105.87	152.71	143.58	Y
*HMGS*	1	Glyur000195s00012841.1	Cytoplasm	2.35	2.81	7.56	173.37	280.56	276.05	Y
*HMGR*	4	Glyur000037s00002618.1	Endoplasmic Reticulum	32.41	57.49	47.67	92.79	193.02	173.28	Y
		Glyur000135s00007303.1	Endoplasmic Reticulum	11.95	15.05	16.18	66.08	35.39	39.91	Y
		Glyur000203s00012900.1	Endoplasmic Reticulum	6.69	12.23	17.49	26.99	42.77	38.50	Y
		Glyur000682s00024324.1	Endoplasmic Reticulum	13.21	16.87	18.17	137.41	182.13	172.25	Y
*MK*	1	Glyur000069s00004081.1	Extracell	5.15	6.75	6.73	9.61	6.62	6.40	N
*PMK*	1	Glyur000343s00025703.1	Cytoplasm	16.23	9.45	10.92	5.40	13.19	13.26	N
*MPD*	1	Glyur000002s00000233.1	Cell Membrane and Extracell	7.42	9.09	9.11	30.96	41.07	42.35	Y
*FPPS*	1	Glyur000088s00007722.1	Endoplasmic Reticulum and Nucleus	5.14	6.08	11.84	35.58	40.90	40.94	Y
*SQS*	2	Glyur000017s00002413.1	Extracell	21.40	19.05	24.26	18.33	48.71	46.67	Y
		Glyur000089s00008825.1	Endoplasmic Reticulum.	3.70	3.44	4.82	16.81	43.09	38.43	Y
*β-AS*	1	Glyur001733s00027628.1	Extracell and Nucleus	0.97	2.28	10.35	45.99	189.49	162.89	Y
*CYP88D6*	2	Glyur000561s00023443.1	Cytoplasm Endoplasmic Reticulum and Nucleus	0.04	0.39	0.91	12.41	21.66	17.30	Y
		Glyur000561s00023451.1	Endoplasmic Reticulum.	0.25	0.64	2.67	183.15	327.26	297.26	Y
*CYP72A154*	2	Glyur000890s00019071.1	Endoplasmic Reticulum.	6.92	7.14	6.31	118.31	181.82	175.76	Y
		Glyur001936s00032203.1	Extracell.	10.00	1.51	7.57	103.78	167.71	142.77	Y
*GUCPR1*	1	Glyur000294s00011848.1	Endoplasmic Reticulum.	38.03	46.50	39.00	16.14	25.88	25.71	Y

^1^ “L1, L2, L3” and “R1, R2, R3” represent 3 leaves and roots samples, respectively. “Y” means yes. “N” means no.

**Table 2 ijms-21-03101-t002:** Analysis of the R2R3-MYB binding site.

Gene ID	Structural Gene Name	Predicted Binding Site Upstream ATG (bp)
Glyur000218s00011642.1	*AACT*	−1934
Glyur000682s00024324.1	*HMGR*	−1544
Glyur000069s00004081.1	*MK*	−1875
Glyur000002s00000233.1	*MPD*	−1748
Glyur000089s00008825.1	*SQS1*	−696
Glyur000561s00023451.1	*CYP88D6*	−340 and −347

**Table 3 ijms-21-03101-t003:** AS occurred in structural genes of GA.

Gene ID	Gene Name	Event Type	Chrom	Event Pattern
Glyur000294s00011848	*CPR*	TTS	Scaffold00294	18154
		TSS	Scaffold00294	10567
Glyur000890s00019071	*CYP 72A154*	TTS	Scaffold00890	8048
		XIR_ON	Scaffold00890	8315–8798
		TSS	Scaffold00890	10471
Glyur001936s00032203	*CYP 72A154*	TTS	Scaffold01936	49804
		TSS	Scaffold01936	52193
Glyur000218s00011642	*AACT*	TTS	Scaffold00218	225653
		TSS	Scaffold00218	229590
Glyur000195s00012841	*HMGS*	TTS	Scaffold00195	66634
		TSS	Scaffold00195	71499
Glyur000682s00024324	*HMGR*	TTS	Scaffold00682	96974
		TSS	Scaffold00682	95210
Glyur000037s00002618		TTS	Scaffold00037	147378
		TSS	Scaffold00037	149004
Glyur000135s00007303		TTS	Scaffold00135	140846
		TSS	Scaffold00135	142733
Glyur000203s00012900		TTS	Scaffold00203	184607
		TSS	Scaffold00203	186414
Glyur000069s00004081	*MK*	TTS	Scaffold00069	351209
		TSS	Scaffold00069	345493
		IR_OFF	Scaffold00069	347728–347912,348276–348369
		IR_ON	Scaffold00069	347728–348369
Glyur000343s00025703	*PMK*	TTS	Scaffold00343	178004
		TSS	Scaffold00343	171667
Glyur000002s00000233	*MPD*	TTS	Scaffold00002	176640
		TSS	Scaffold00002	173112
Glyur000088s00007722	*FPPS*	TTS	Scaffold00088	174563
		TSS	Scaffold00088	178062
Glyur000017s00002413	*SQS2*	TTS	Scaffold00017	277461
		TTS	Scaffold00017	277670
		TTS	Scaffold00017	277888
		TSS	Scaffold00017	271392
Glyur000089s00008825	*SQS1*	SKIP_OFF	Scaffold00089	317631–318581
		SKIP_ON	Scaffold00089	317631,318018–318093,318581
		TTS	Scaffold00089	314669
		TSS	Scaffold00089	319057
		IR_OFF	Scaffold00089	315038–315129,315471–315559
		IR_ON	Scaffold00089	315038–315559
Glyur001733s00027628	*β-AS*	TTS	Scaffold01733	36913
		TSS	Scaffold01733	44011
Glyur000561s00023451	*CYP88D6*	TTS	Scaffold00561	103653
		TTS	Scaffold00561	139473
		TSS	Scaffold00561	143458

**Table 4 ijms-21-03101-t004:** miRNAs target structural genes.

miRNA Sequence	miRNA Name	Target Gene ID	Target Annotation	Expectation
AGCAAAAAAGUACAUGUAUCUGGA	miRNA-n217c	Glyur000069s00004081.1	*MK*	3
AGCAAAAAAGUACAUGUAUCUGGA	miRNA-n217b	Glyur000069s00004081.1	*MK*	3
AGCAAAAAAGUACAUGUAUCUGGA	miRNA-n217a	Glyur000069s00004081.1	*MK*	3
AGCAAAAAAGUACAUGUAUCUGGA	miRNA-n217e	Glyur000069s00004081.1	*MK*	3
AGCAAAAAAGUACAUGUAUCUGGA	miRNA-n217d	Glyur000069s00004081.1	*MK*	3
AAGAAAAAAAGUACAUGCAUCUGA	miRNA-n217f	Glyur000069s00004081.1	*MK*	2
AGAAAAAAAAUACAUGUAUCUGGA	miRNA-n217i	Glyur000069s00004081.1	*MK*	1
AGAAAAAAAAUACAUGUAUCUGGA	miRNA-n217h	Glyur000069s00004081.1	*MK*	1
AGAAAAAAAAUACAUGUAUCUGGA	miRNA-n217g	Glyur000069s00004081.1	*MK*	1
AUUCAGCUGGGGCUCACAUCCACC	miRNA-n244	Glyur001733s00027628.1	*β-AS*	3
CAAAUACACUGCAGAACUAAUGU	miRNA-n69	Glyur000682s00024324.1	*HMGR*	3
UCAAAGGGGGGUGUGGAGUAG	miR2119	Glyur000294s00011848.1	*CPR1*	2.5

## Data Availability

All the raw data of the transcriptome and small RNAs are deposited in the NCBI short read archive (SRA) under the accession numbers PRJNA555398 and PRJNA555703, respectively.

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
