# Peer review of "Integrated Analysis of mRNA and microRNA Elucidates the Regulation of Glycyrrhizic Acid Biosynthesis in Glycyrrhiza uralensis Fisch"

_ijms, 2020, doi:10.3390/ijms21093101_

Round 1
Reviewer 1 Report
The article entitled “Integrated analysis of mRNA and microRNA elucidates the regulation of glycyrrhizic acid biosynthesis in Glycyrrhiza uralensis Fisch” by Yanni et al. is interesting and worth to read. The authors attempted to elucidate the gene expression and regulation of GA biosynthesis in G. uralensis. However, before consider to publish the authors need to perform some minor revisions.
- This article cite more than 100 references. Please try to avoid un-necessary citations from the text
- Abbreviations are not properly mentioned in the text and lot of typos. Please correct them all
EX- The primary bioactive compound in Gancao is GA, ….. (Line 37-38) This should be “The primary bioactive compound in Gancao is glycyrrhizic acid (GA), ”
EX- PAL, C4H, CHS, 4CL and CHR in soybean……=> PAL, C4H, CHS, 4CL, and CHR in soybean
(Line 24, 40, 70, 79, 116, 442, 402, 447, 491,++++)
- Remove subtitles from discussion
Author Response
- This article cite more than 100 references. Please try to avoid un-necessary citations from the text
Response: Thanks for the suggestion. I have deleted the un-necessary citations from the text.
- Abbreviations are not properly mentioned in the text and lot of typos. Please correct them all
EX- The primary bioactive compound in Gancao is GA, ….. (Line 37-38) This should be “The primary bioactive compound in Gancao is glycyrrhizic acid (GA), ”EX- PAL, C4H, CHS, 4CL and CHR in soybean……=> PAL, C4H, CHS, 4CL, and CHR in soybean(Line 24, 40, 70, 79, 116, 442, 402, 447, 491,++++)
Response:
- I have corrected “GA” to “glycyrrhizic acid (GA)” (Line 38).
- Line 24's TF, TR, PK and GA have written their full names on Line 15, 20, 21.
- I have corrected “PAL, C4H, CHS, 4CL and CHR in soybean” to “phenylalanine ammonia-lyase (PAL), cinnamate 4–hydroxylase (C4H), chalcone synthase (CHS), 4-coumarate: coenzyme A ligase (4CL), chalcone reductase (CHR) in soybean” (Line 69-71).
- I have corrected “SnRK2” to “SNF1-related protein kinase 2 (SnRK2)”and corrected “PP2C” to “phosphatase 2C (PP2C)” (Line 80).
- I have corrected “IFS” to “isoflavone synthase (IFS)” (Line 82).
- I have corrected “AtCPK1-Ca ” to “Arabidopsis thaliana calcium-dependent protein kinase 1(AtCPK1)”; “HID” to “hydroxyisoflavanone dehydratase (HID)”;” IDMAT” to “dimethylallyltransferase (IDMAT)”; “C4-DMAT”to “coumestrol 4-dimethylallyltransferase (C4-DMAT)” (Line 83-85).
- I have corrected “with a stem-loop structure followed by action of Dicer, SE,HYL1, HEN1, HASTY, AGO, STA1 and RNA binding protein” to “with a stem-loop structure. After the action of DICER-LIKE1(DCL1), SERRATE (SE),HYPONASTIC LEAVES1 (HYL1), HUA ENHANCER1 (HEN1), HASTY(HST), ARGONAUTE1(AGO1), STABILIZED1 (STA1) and RNA binding protein”(Line 88-91).
8.I have corrected “COG, GO, KEGG, KOG, Pfam, Swiss-Prot, eggNOG and NR databases” to “Cluster of Orthologous Groups of proteins (COG), Gene Ontology (GO), Kyoto Encyclopedia of Genes and Genomes (KEGG), euKaryotic Ortholog Groups (KOG),Protein family (Pfam), Swiss-Prot protein sequence (Swiss-Prot), Nonsupervised Orthologous Groups(eggNOG), NCBI nonredundant protein sequences (NR) databases”(Line 125-128)
9. I have corrected “23 transporters to be located in the endoplasmic reticulum.” to “23 transporters to be located in the endoplasmic reticulum (ER).” (Line221)
10.I have corrected “CESA” to “Cellulose Synthase (CESA)” (Line 382)
11.I have corrected “GH2” to “gold hull and internode2 (GH2)” (Line 393). 3.Remove subtitles from discussion
Response: Thanks for the suggestion. I have removed subtitles from discussion.

Reviewer 2 Report
Comments are in the attached file

Author Response
Comments of reviewer 2
The authors are reporting an extensive transcriptome analysis of leaves (low GA) and
roots (high GA) of Chinese licorice and identify differentially expressed genes (DEGs)
between the 2 organs, as a mean to fish new genes involved in the regulation of the
biosynthetic pathway of GA. Several bionformatic analysis are presented. In
particular the authors apply co-regulation analysis of expression of the main
biosynthetic genes with TF, TR, PK and transporters. The results point to the
identification of candidate genes involved in the regulation and transport of GA
which is relevant. MYB binding sites are also discovered in the promoter of several
structural genes, being MYB TFs a highly represented category in the TFs
co-expressed with the structural genes themselves.
The authors also present a comprehensive small RNAome analysis with a focus on
target prediction of miRNA and highlights on the presence of DEGs among target
genes; in particular miRNA targeting structural genes and co-regulated genes (TF,
TR,PK and transporters) are also reported.
Concerning transporters the authors are able to restrict by ER localization their
candidates to a reasonable numer, 23, to be analysed by functional strategies.
Among TFs, 3 MYBs are pinpointed as co-expressed with structural genes, because 6
structural genes have MYB binding sites in their promoters, the 3 MYBs are robust
candidates for further analysis. For TR and PK genes a relatively high number of
candidates is maintained. miR156 is suggested as a possible candidate because it
potentially targets 15 TFs that are co-expressed with structural genes.
In brief after transcriptomics and extensive bioinformatic analysis a numeber of
candidate gene potentially involved in regulation of GA biosynthetic genes are
revealed which is a valuable contribution to functional studies for their validation
and also for future use in biotechnolgical applications.
There is a partial problem of originality as a similar work that should be cited
Ramilowski et al 2013 is published in Plant Cell Phys. The latter paper reports a
transcriptomic analysis of low and high content of glycyrrhizin tissues of licorice and
search for P450s genes and trnasporters for saponins. To my opinion the current
authors are presenting a wider range of candidate genes for GA synthesis and
regulation, not only transporters but also TF,TR and PK also miRNA and in particular
they narrow the number of candidates for some class of genes.
Another very recent paper Gao et al 2020 (published in BMCL) which can also be
cited also reports a transcriptomics analysis in G. glabra and findings on DEGs from
tissues having a different content of glycyrrhizic acid; their analysis related to the
genes for terpenoid biosynthesis is not as deep as the one proposed by the current
authors.
Response: Thanks for the suggestion. The work of Ramilowski et al was cited the in line 48 of the main text.
Therefore I think the current dataset is valuable for publication after minor
corrections on language style and minor mistakes.
Below some minor corrections are reported:
52 skeleton
and modification of the skeleton [6]. For the first stage, hydroxyl
methylglutaryl-CoA
53 reductase
(HMGR) is the first rate-limiting enzyme that catalyzes transformation of
HMG-CoA to MVA
Response: I have corrected “catalyzes HMG-CoA to MVA” to “catalyzes transformation of HMG-CoA to MVA” (Line53).
54 Squalenesynthase (SQS), a key enzyme for the flow of carbon to triterpenes, catalyzes the
synthesis of FPP to
55 squalene [8]. Then, β-AS transforms 2,3-oxidosqualene to β-amyrin by two consecutive
oxidation
56 reactions [9].
Response: I have corrected “transfers” to “transforms” (Line55).
59 oxidation to balance the redox environment in the body (in the cell)
Response: I have corrected “in the body” to “in the cell” (Line 59).
Plant miRNAs are nonprotein-coding RNAs measuring approximately 21-24 nt, which are
84 usually derived from a primary transcript with a stem-loop structure. Afer the action of
Dicer, SE,
85 HYL1,
HEN1, HASTY, AGO, STA1 and RNA binding protein mature miRNAs are
produced [35-37]
Response: I have corrected “with a stem-loop structure followed by action of Dicer, SE,” to “with a stem-loop structure. After the action of DICER-LIKE1(DCL1), SERRATE (SE),”(Line88-89).
90 In this study, we constructed libraries of the transcriptome and small RNAome using leaf
and
91 root
samples of G. uralensis for different GA contents in these two tissues (they mean
that the 2 tissues have different GA content therefore they performed trnascriptomics, The
sentence has to be improved)
Response: I have corrected the sentence as “In this study, because roots and leaves in G. uralensis have different GA content, we performed transcriptomics and small RNAomics analysis by using these two tissues”(Line96-97).
Tab.1 L1 L2 L3 and R1 R2 R3 should correspond to the 3 leaf and root samples
respectively, it should be mentioned in the table caption.
275 As shown in Fig. 6C (Fig.6 is mentioned before Fig.5 which ismentioned at 304 for the
first time, I thinks they should be re-numbered), the groups of 21 nt and 24 nt sRNAs had
the highest proportions of U and
276 A
nucleotides at the fir
Response: Thanks for the suggestion. “As shown in Fig. 6C” description error, we have changed it to " As shown in Figure S1C "
364 In this study, we integrated mRNA and microRNA (transcriptional analysis or
something similar) to reveal the expression and regulation of
365 GA
biosynthesis in G. uralensis.
Response: Thanks for the suggestion. I have corrected the sentence as “In this study, we performed transcriptional analysis to reveal the expression and regulation of GA biosynthesis in G. uralensis” (Lin376-377).
Usually, the longest transcript was
376 chosen
Entire
as a component for synthetic biology because it could encode proteins with
Commented [M1]: Component for synthetic biology is
unclear
377 domains.
Plant Materials and RNA Extraction
419 Two-year-old
- uralensisplants (accession type,cultivar some reference on the origin
of the material is necessary) were grown in the experimental field of Beijing Medicinal Plant
Response: Thanks for the suggestion. I have added “B11”, a wild accession (unpublished) after the plants.

Reviewer 3 Report
In this study, Li and colleagues reported RNA-Seq analysis of three leaf and three root samples of Glycyrrhiza uralensis (Licorice), which is an important herb due to its production of glycyrrhizic acid (GA). Based on their RNA-Seq data, the authors performed bioinformatics analysis in an attempt to “elucidate the gene expression and regulation of GA biosynthesis in G. uralensis”. Although this research goal is of potential pharmacological value, there appears a severe concern about their experimental design. Particularly, the majority of the data shown in this manuscript are based solely on the comparison of gene expression in leaves vs roots, because the authors reasoned that “the content of GA is higher in roots than in leaves”. However, the so-called “differentially-expressed genes (DEGs)” presented in this manuscript are simply those genes that are not expressed at the similar levels in leaves vs roots. There is not necessarily any relationship between these DEGs and a potential function in GA biosynthesis. This transcriptomics approach is meaningful only when comparing the same tissue type from two varieties of G. uralensis with different GA content (if any). Besides the concerned DEGs, although the prediction analysis such as transcription factor binding sites and potential targets of miRNAs may be interesting, the findings are premature without any experimental verifications.
Author Response
Comments and Suggestions for Authors
In this study, Li and colleagues reported RNA-Seq analysis of three leaf and three root samples of Glycyrrhiza uralensis (Licorice), which is an important herb due to its production of glycyrrhizic acid (GA). Based on their RNA-Seq data, the authors performed bioinformatics analysis in an attempt to “elucidate the gene expression and regulation of GA biosynthesis in G. uralensis”. Although this research goal is of potential pharmacological value, there appears a severe concern about their experimental design. Particularly, the majority of the data shown in this manuscript are based solely on the comparison of gene expression in leaves vs roots, because the authors reasoned that “the content of GA is higher in roots than in leaves”. However, the so-called “differentially-expressed genes (DEGs)” presented in this manuscript are simply those genes that are not expressed at the similar levels in leaves vs roots. There is not necessarily any relationship between these DEGs and a potential function in GA biosynthesis. This transcriptomics approach is meaningful only when comparing the same tissue type from two varieties of G. uralensis with different GA content (if any). Besides the concerned DEGs, although the prediction analysis such as transcription factor binding sites and potential targets of miRNAs may be interesting, the findings are premature without any experimental verifications.
Response: Thanks for the suggestions very much.
In this study, we attempted to elucidate the gene expression and regulation of GA biosynthesis in G. uralensis. Compared with roots, most of the GA biosynthetic pathway genes are differentially expressed in leaves, because the content of GA is higher in roots than in leaves. Then we apply co-expression analysis of the main biosynthetic genes with TF, TR, PK and transporter. The results point to the identification of candidate genes involved in the regulation and transport of GA which is relevant.
In the future, we will compare the same tissue type from two varieties of G. uralensis with different GA content to improve our work to mine genes potentially involved in GA biosynthesis.
We will also carry out further experiments to validate the transcription factor binding sites and potential targets of miRNAs, etc.

Round 2
Reviewer 3 Report
No revision has been made in this manuscript to support a change in this reviewer's opinion.